# LIDAR-Inertial Real-Time State Estimator with Rod-Shaped and Planar Feature

**Hong Liu** [1,2] , **Shuguo Pan** [1,2,*], **Wang Gao** [1,2], **Chun Ma** [1,2] , **Fengshuo Jia** [1,2] and **Xinyu Lu** [1,2]

1   School of Instrument Science and Engineering, Southeast University, Nanjing 210096, China
2   Key Laboratory of Micro-Inertial Instrument and Advanced Navigation Technology, Southeast University, Nanjing 210096, China
*   Correspondence: psg@seu.edu.cn

**Abstract:** State estimation and mapping based on Light Detection and Ranging (LIDAR) are important for autonomous systems. Point cloud registration is a crucial module affecting the accuracy and real-time performance of LIDAR simultaneous localization and mapping (SLAM). In this paper, a novel point cloud feature selection for LIDAR-inertial tightly coupled systems is proposed. In the front-end, a point cloud registration is carried out after marking rod-shaped and planar feature information which is different from the existing LIDAR and inertial measurement unit (IMU) integration scheme. This preprocessing method subsequently reduces the outliers. IMU pre-integration outputs high-frequency result and is used to provide the initial value for LIDAR solution. In the scan-to-map module, a computationally efficient graph optimization framework is applied. Moreover, the LIDAR odometry further constrains the IMU states. In the back-end, the optimization based on sliding-window incorporates the LIDAR-inertial measurement and loop closure global constraints to reduce the cumulative error. Combining the front-end and back-end, we propose the low drift and high real-time LIDAR-inertial positioning system. Furthermore, we conducted an exhaustive comparison in open data sequences and real-word experiments. The proposed system outperforms much higher positioning accuracy than the state-of-the-art methods in various scenarios. Compared with the LIO-SAM, the absolute trajectory error (ATE) average RMSE (Root Mean Square Error) in this study increases by 64.45% in M2DGR street dataset (street_01, 04, 07, 10) and 24.85% in our actual scene datasets. In the most time-consuming mapping module of each system, our system runtime can also be significantly reduced due to the front-end preprocessing and back-end graph model.

**Keywords:** tightly-coupled integration; LIDAR-inertial SLAM; rod-shaped and planar feature; sliding-window; graph optimization framework

## 1. Introduction

Accurate and reliable state estimation is a fundamental requirement of mobile robot and automatic driving. In urban environments, indoor environments and other complex scenes, it is difficult to achieve a high precision of positioning requirements with the traditional GNSS/INS integrated.

In recent years, visual/LIDAR simultaneous localization and mapping have made certain developments. On the one hand, visual slam can achieve six degrees-of-freedom state estimation just by camera, but it is seriously affected by the illumination and low texture feature [1]. On the other hand, the laser sensor directly obtains depth information and has high resolution, which can also work at night and achieve accurate pose estimation. Therefore, this research mainly focuses on LIDAR simultaneous localization and mapping.

LIDAR odometry and mapping (LOAM) [2] is an earlier proposed LIDAR slam algorithm. Iterative ICP algorithm is a common method for point cloud matching, which is time-consuming for registration, and it is easy to fall into a local minimum [3]. LOAM replaces ICP with point-to-line and point-to-plane matching. It consists of two subsystems.

The odometry system performs point to line/plane feature matching to calculate pose between scans. The features of line and plane are judged according to point curvature. At the same time, LOAM effectively eliminates the unreliable parallel points and occlusion points. And it performs the distortion compensation by motion interpolation. The mapping system performs scan to map matching and runs at lower frequency, which can perform higher accuracy state estimation. By combining these two systems, LOAM achieves low drift and low-computational complexity, which has been ranked as the top in the LIDAR based method on the KITTI odometry benchmark site [4]. However, LOAM still has some flaws; its point cloud is stored in global voxel. Without key frame selection, it is difficult to integrate observation information of other sensors and perform global optimization.

F-LOAM adopts a two-stage distortion compensation method to reduce the computational cost and improve the real-time performance [5], but there are still no global optimization methods such as loop closure, resulting in large cumulative errors over a long period of time. Liu et al. propose a method based on deep learning for extracting feature points and obtaining the descriptors in LIDAR odometry. It also adopts the two-step state estimation for long distance experiment, which has a good performance for LIDAR of various resolutions [6]. V-LOAM introduces the visual odometry as the front-end of the laser odometry, further improving the accuracy of slam [7]. HDL_GRAPH_SLAM [8] is an algorithm that can fuse LIDAR, IMU and GNSS sensors, but the scan registration accuracy is low which is based on NDT [9]. It is also prone to drift in non-plane because of the flat ground constraint. LeGo-LOAM implements point cloud segmentation to reduce the number of features, and two-step registration provides the initial value for LIDAR mapping module. LeGo-LOAM firstly covers the key frame selection and loop detection [10]. However, there is obvious drift in the large scene testing experiment and the IMU is only used to remove distortion. LIO-mapping [11] is a joint state estimation problem based on the ideas of LOAM and VINS-Mono [12]. The front-end vision part is replaced by the LIDAR front-end for feature extraction and state estimation. However, the optimization problem is too large to be real-time, which makes it hard to apply in a mobile device. LINS is a tightly coupled LIDAR-inertial odometry (LIO) system based on the filter method [13]. The iterative error state Kalman Filter is used to correct the state estimation of the robot, but there is still a problem that the robot will drift when it runs for a long time without global constraints. LiLi-OM puts forward an adaptive keyframe selection for both solid-state and traditional LIDAR. It also introduces a metric weighting function during sensor fusion [14]. However, lacking a point cloud processing, the system stability is inadequate. LIO-SAM [15] is also a tightly-coupled LIO system, which is based on the incremental smoothing and mapping framework iSAM2 [16]. In addition, the loop closure factor and GPS factor can be added to the global optimization factor graph. In spite of this, its IMU constraints do not enter the factor graph optimization system, which may result in loss of constraint information between IMU and LIDAR measurements. In the actual scene test, LIO-SAM will appear at unstable states such as point cloud matching errors, especially when the carrier movement is in a large scene. Zhang et al. proposed the LIDAR-inertial odometry with an adaptive covariance estimation algorithm which is based on loosely-coupled method. It achieves better result compared to the tightly-coupled method [17].

In short, the existing LIDAR slam algorithms are mainly for small scenes. But for the complex scene or the great motion change, they are prone to cumulative errors and poor robustness.

Meanwhile, the processing of point cloud data affects the accuracy of point cloud registration for LIDAR slam. Douillard et al. introduced a method which jointly determines the ground and individual objects on the ground in three-dimensional space, including overhanging structures, but it requires a large amount of computation time, limiting online applications [17]. B. Douillard et al. proposed a priori ground extraction way. Segmentation of dense 3D data is optimized via a simple yet efficient voxel of the space. This approach provides near-real-time performance, but is not sufficient for real-time positioning scenarios [18]. M. Himmelsbach et al. proposed that 3D point clouds are

projected onto 2D grids on the ground plane, and then point clouds were segmented on the occupied grids [19]. The algorithm has fast speed and is suitable for online segmentation. However, the method tends to result in weak segmentation. When two objects are relatively close to each other, it is prone to misrecognition, especially in the z-axis direction. In 2019, Seungcheol Park et al. proposed Curved-Voxel Clustering. The point cloud coordinates are converted from cartesian coordinates to spherical coordinates, and each point cloud is assigned to the voxel in the corresponding spherical coordinate system. Hash tables establish associations between indexes and points. When clustering, lookup is implemented using the hash tables [20]. Chen et al. use IMU to assist the point cloud registration and introduce the inertial error model for mobile laser scanning, which could effectively reduce the error with low time cost [21].

This paper mainly aims to improve the accuracy of LIDAR point cloud registration under the condition of real-time positioning, so as to ensure the robustness of the system. The contributions of this paper are summarized as follows:

1.  A quick and effective feature extraction method is proposed. Due to the information of rod-shaped and planar feature, edge points and surface points are extracted reasonably to calculate curvature with the low computational cost.
2.  IMU pre-integration is used to provide the initial value for LIDAR odometry, and the LIDAR odometry further constrains the pre-integrated IMU states.
3.  A graph optimization model is used to solve the scan-to-map module, which greatly improves the speed of the traditional algorithm. Another graph optimization model is used to globally optimize the pre-integrated IMU measurements residuals, inter-frame matching residuals and loop residuals, which improves the accuracy and stability of LIO system effectively.

## 2. System Overview

The overall framework of this system is shown in the Figure 1. LIDAR and IMU measurements are the inputs for the system.

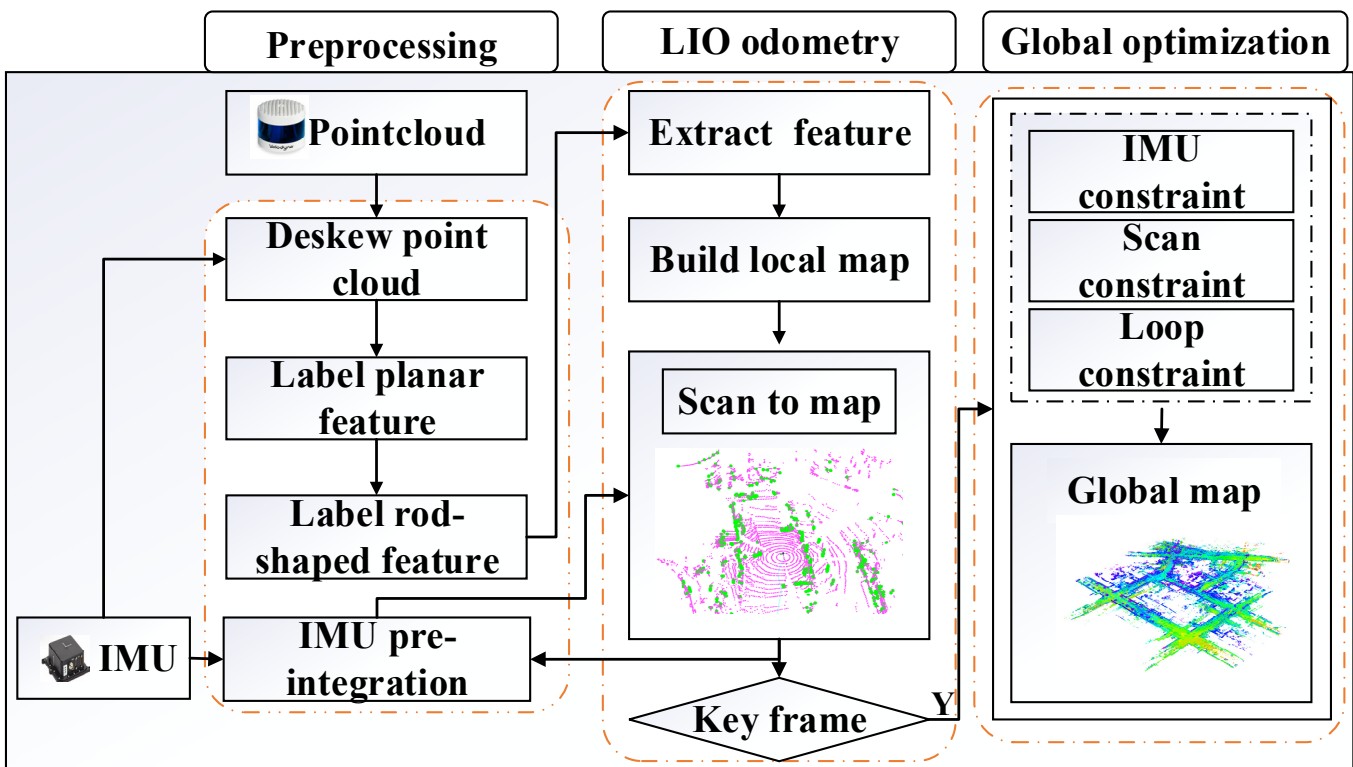

**Figure 1.** Overall framework of our LIO system.

The system can be divided into three parts.

First, *the preprocessing module*, the raw point clouds are de-skewed using gyroscope data and IMU pre-integration value. Current scan's point cloud is projected to the 2D image. The depth characteristic value is used to remove the outlier points. Image is used to segment the planar feature and cluster the rod-shaped feature information.

Then, *the LIO odometry module*, IMU pre-integration results are used to estimate motion pose. The scan-to-map between current frame and local map is performed. In the scan-to-map module, we introduce the graph optimization model which can enhance the speed and accuracy of the solution, and a sliding window-based way is applied to update and maintain the local map.

At last, *the global optimization module*, if the current frame is judged to be a keyframe, LIDAR scan-to-scan residuals, pre-integrated IMU residuals and loop residuals are optimized via the slide window optimization. Information of marginalization is used for prior constraints. Loop closure is detected and performed in an effective way, which is beneficial to reduce cumulative error.

According to this system, we get the 6-DOF pose estimation and a real-time updated global map. Exhaustive comparisons have been conducted to prove the superiority of our system.

We define notations and frame definitions throughout the article. $(\cdot)^W$ is considered as world frame. In the LIO system, the origin of the world coordinate is identified as the first LIDAR frame $(\cdot)^B$ is the body frame and $(\cdot)^L$ is the LIDAR frame. Rotation is represented by rotation matrices $R$ and quaternions $q$. So $R_W^B$ and $q_W^B$ is the rotation from world frame to body frame, and $p_W^B$ is the translation. $\otimes$ is defined as the multiplication between two quaternions.

## 3. The Preprocessing Module

In this study, the current LIDAR point cloud is projected onto the current 2D image grid, which is represented by a matrix. The horizontal index unit of moment frame is the horizontal resolution of each frame, and the vertical unit is the vertical resolution. For example, the size of the projected image matrix of 16-line LIDAR is $16 \times 1800$. The value of the image grid stores the depth of each point, and the points will be removed if there is an outlier value. The operating point cloud data on the basis of two-dimensional images can significantly improve the computing speed.

After this process, reliable estimation of LIDAR´s per scan is a necessary prerequisite. In this paper, IMU pre-integration is used to obtain the relative translational motion at the beginning and the end of each scan. Based on this method, point cloud distortion can be eliminated. In the meantime, the raw point clouds from each scan are rotationally de-skewed using gyroscope data.

### 3.1. Label Planar Feature Information

The sensor carrier is moving on the ground and the LIDAR is mounted horizontally. The ground is observed with the beams below. We can get a rough but fast estimate of the plane from the number of rows of the image matrix. In the estimation plane, accurate ground points can be marked by judging the angle of each point to the ground.

$$a = \tan^{-1}\frac{dz}{\sqrt{(dx)^2 + (dy)^2}} = \frac{OP_2}{OP_1} \tag{1}$$

As shown in the Figure 2, $P_1$ and $P_2$ are two laser beams reflection points. The angle *a* corresponding to the points of adjacent laser beams should have a small value if there is no barrier. Points on the ground can be marked according to the size of the included angle value. *Dx*, *dy* and *dz* represent the differences of the two laser beams in the three directions, respectively.

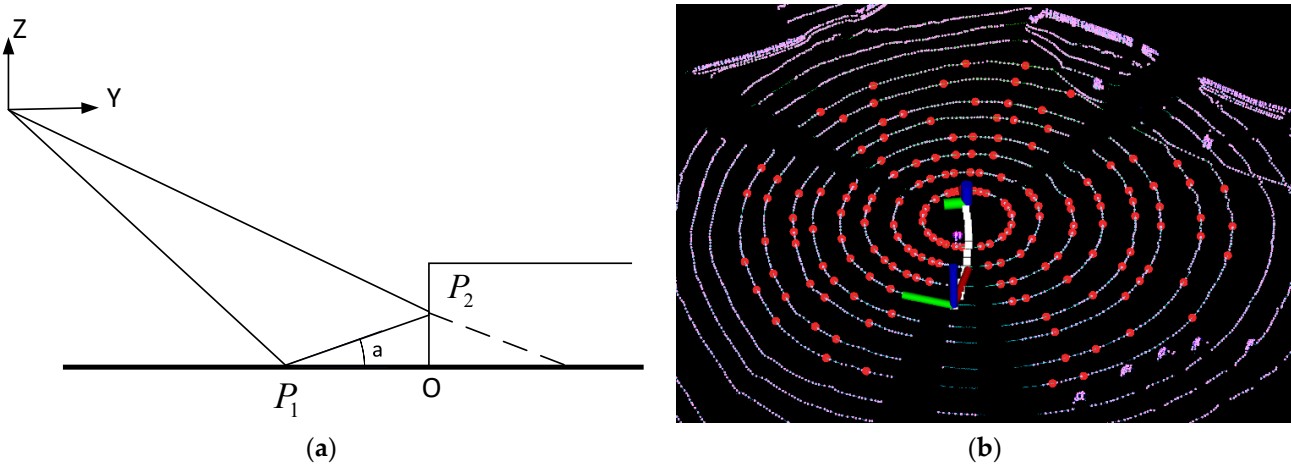

**Figure 2.** Illustration of the planar feature information (**a**) The angle between the lines for judging the surface; (**b**) Red points are the planar feature in the scan.

*3.2. Label Rod-Shape Feature Information*

Figure 3 is the top view; OC is the measurement of the first beam and OD is the second.

$$B = \tan^{-1}\frac{r_2\sin\alpha}{r_1 - r_2\cos\alpha} = \frac{MD}{OM} \tag{2}$$

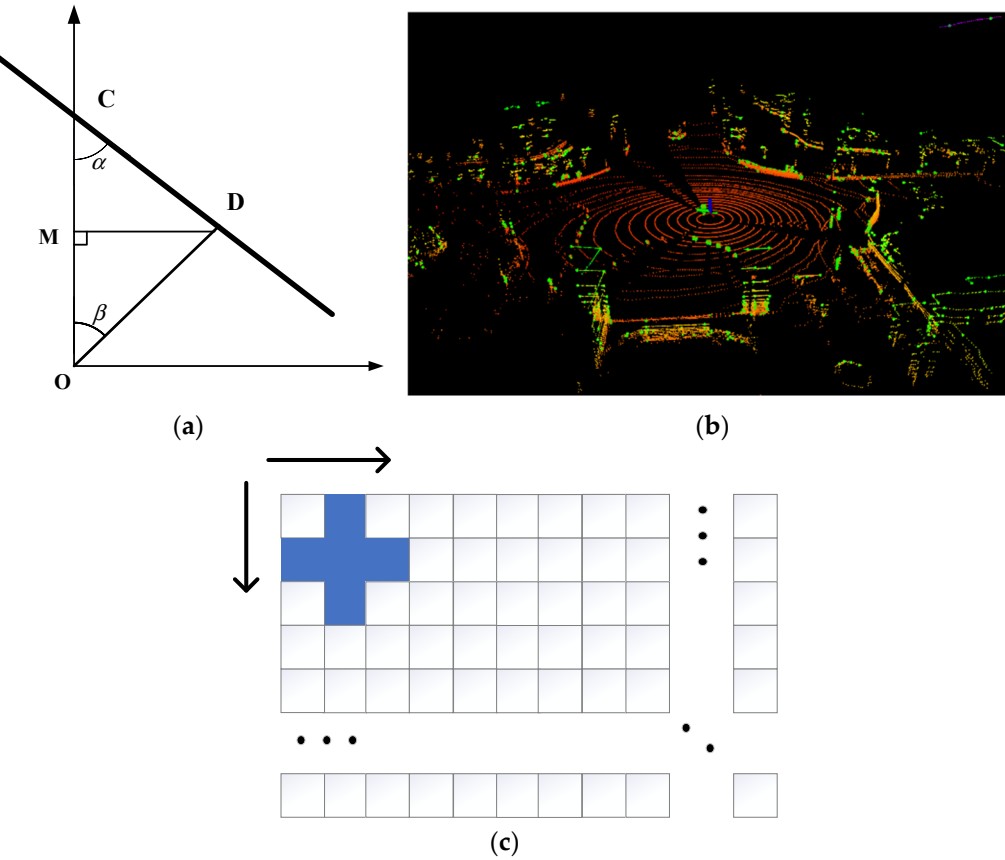

**Figure 3.** Illustration of the rod-shape feature cluster (**a**) The fast and accurate method for segmentation; (**b**) Green points are the edge feature in the scan; (**c**) The BFS search method in current scan.

In this formula, $r_1$ and $r_2$ are the depth values measured by the two beams. $A$ is the angle between the two laser beams. For example, the angle is 0.2° in x direction and 2° in y direction in Velodyne16.

If C and D are on different objects, the angle $\beta$ between OC and OD will be greater than a certain threshold. In this work, we set this threshold at 20°. According to this method, we can cluster the same cluster objects quickly and accurately.

The point cloud of our two-dimensional image model can be traversed quickly. We traversed each pixel in the 2D image and calculated the included angle for four points around each pixel. And each pixel is searched by BFS algorithm [22].

As shown in the Figure 3c, the surrounding points consist of the left, right, lower and top pixels. If the angle between the surrounding points is smaller than the threshold, we decided they are the same object. If a point is marked, then it will be skipped. So, the algorithm complexity is $\Theta(N)$, where N is the number of image pixels.

For the subsequent processing of point cloud, the influence of disorderly points and inaccurate points can be avoided. For example, when the carrier is driving, leaves, small objects, grass and weeds can be removed. These are difficult to observe through two consecutive frames of scanning, which are the main factors affecting the pose solution.

### 3.3. Feature Extraction

Through the segmentation and clustering of planar and rod-shaped information in the previous steps, the extraction of edge point and planar point are carried out in the rod-shape and planar feature information. Curvature is defined as follows:

$$c = \frac{1}{|S| \bullet ||\mathrm{Pr}_i^L||} || \sum_{j \in S, j \neq i} (\mathrm{Pr}_j^L - \mathrm{Pr}_i^L)|| \tag{3}$$

In this formula, $S$ is the set of continuous points from the same row of the 2D image. Pr is the point range. In this work, $S$ is set to 10. And $c$ is the curvature value.

Similar to LOAM, the depth information of each point is used to eliminate parallel points and occluded points, which have certain influence on the subsequent solution. In the feature extraction process, each frame is divided into 6 sub-images, which has a resolution of $16 \times 300$. Edge points and planar points are extracted from each sub-image, which are determined according to threshold $c_{th}$ and $p_{th}$. In this work, $c_{th}$ and $p_{th}$ are chosen to be 1 and 0.1. The edge point set and planar point set extracted from k frame are $\varepsilon_k$ and $s_k$.

After this process, we selectively obtain stable features and reduce the calculation pressure at the back-end. At the same time, this step improves the reliability of front-end scan registration.

## 4. The LIO Odometry Module

### 4.1. IMU Pre-Integration

LIDAR and IMU work in different frequencies. Usually, the LIDAR is 10 to 30 Hz and the IMU is 100 to 500 Hz. The pre-integration integrates the IMU measurement values between each adjacent frame of LIDAR, and adopts a value to express it. Through this step, we can get the output of the two sensors at the same frequency

The measurements of IMU include angular velocity $\widetilde{\omega}_B(t)$ and acceleration $\widetilde{a}_B(t)$. The measured values are all under the B coordinate system. And the measurement equation can be modeled as:

$$\widetilde{\omega}_B(t) = \omega_B(t) + b^\omega(t) + \eta^\omega(t) \tag{4}$$

$$\widetilde{a}_B(t) = R_W^B(t)(a_W(t) - g^W) + b^a(t) + \eta^a(t) \tag{5}$$

The measured values are affected by the slowly varying bias $b(t)$ and white noise $\eta(t)$. The acceleration of gravity in the world system. $g^W = [0, 0, g]^T$ is the gravity vector, which affects the measurement. So, it should be subtracted.

In our work, noise is ignored and the biases are considered the constant during the pre-integration period. The current state value can be obtained based on the derive of pre-integration. Assuming that $j$ is the current frame and $i$ is the last frame. The attitude rotation matrix $R_{WB}^j$, velocity $v_{WB}^j$, and position $p_{WB}^j$ can be expressed as:

$$R_{WB}^j = R_{WB}^i \Delta R_{ij} Exp\left(J_{\Delta R_{ij}}^g \delta b_\omega^i\right) \tag{6}$$

$$v_{WB}^j = v_{WB}^i + g^W \Delta t_{ij} + R_{WB}^i \left(\Delta v_{ij} + J_{\Delta v_{ij}}^\omega \delta b_\omega^i + J_{\Delta v_{ij}}^a \delta b_a^i\right) \tag{7}$$

$$p_{WB}^j = p_{WB}^i + v_{WB}^i \Delta t_{ij} + \frac{1}{2} g^W \Delta t_{ij}^2 + R_{WB}^i \left(\Delta p_{ij} + J_{\Delta p_{ij}}^\omega \delta b_\omega^i + J_{\Delta p_{ij}}^a \delta b_a^i\right) \tag{8}$$

$J$ is for Jacobians, and the details can be found in [23]. The $J_{(\cdot)}^a b_a$ and $J_{(\cdot)}^\omega b_\omega$ means a first-order approximation of the effect of changing the biases to avoid repeated integration. Meanwhile, the terms of pre-integration $\Delta R_{ij}$, $\Delta v_{ij}$ and $\Delta p_{ij}$ can be computed between the frame $i$ and $j$:

$$\Delta R_{ij} = \prod_{k=i}^{j-1} Exp((\omega_B^k - b_\omega^i)\Delta t) \tag{9}$$

$$\Delta v_{ij} = \sum_{k=i}^{j-1} \Delta R_{ik}(a_B^k - b_a^i)\Delta t \tag{10}$$

$$\Delta p_{ij} = \sum_{k=i}^{j-1} (\Delta v_{ik}\Delta t + \frac{1}{2}\Delta R_{ik}(a_B^k - b_a^i)\Delta t^2) \tag{11}$$

### 4.2. Build Local Map

In point cloud registration, the iterative closest point (ICP) algorithm is the most commonly scan registration method. However, as the urban scenes consists of lots of moving targets, ICP registration failure rate is high, which is directly based on raw point data. And ICP is improper for localization and mapping in real time due to its large amount of computation. In LOAM, pose estimation depends on scan-to-scan matching for quick estimation. However, this method is prone to cumulative error. In our work, current scan and local map are matched according to the predict value of IMU pre-integration. Meanwhile, the scan-to-map result is used to correct the IMU accumulative errors.

A local map associated with the current LIDAR frame is constructed. A fixed number of key frame maps within a certain range are constructed by sliding window method. The local map is converted to the W coordinate system. Edge points and planar points of a local map form the voxel map. And the points in local map are down-sampled to eliminate the duplicated features. In order to improve the point cloud matching speed, the feature information of the local map is stored in the data structure of KD-tree [24] for the convenience of subsequent search.

Therefore, this paper adopts the registration method based on feature points. After the feature points with the same type are obtained through preprocess, the graph optimization model is used to iteratively locate for current scan and local map.

### 4.3. Pose Estimation

For each edge point $p_\varepsilon \in \varepsilon_k$, we search for the nearest five points on the local map and calculate the mean and covariance matrices for the five points. When the distribution of points approximates a straight line, one eigenvalue of the covariance matrix will be significantly larger than the rest. In this study, the eigenvector corresponding to the eigenvalue $u_\varepsilon^{sm}$ is the main direction of the line, and $p_\varepsilon^{sm}$ is the geometric center of the five points in the Figure 4. If the line feature satisfies the condition, the distance between the current edge point and the line can be calculated, and the best pose estimation of the

current point in the local map can be obtained by minimizing the distance. The distance calculation formula:

$$f_\varepsilon(p_\varepsilon) = p_n \bullet ((T_k p_\varepsilon - p_\varepsilon^{sm}) \times u_\varepsilon^{sm}) \tag{12}$$

where symbol $\bullet$ is the dot product and $\times$ is the cross product. $p_n$ is the unit vector.

$$p_n = \frac{(T_k p_\varepsilon - p_\varepsilon^{sm}) \times u_\varepsilon^{sm}}{||(T_k p_\varepsilon - p_\varepsilon^{sm}) \times u_\varepsilon^{sm}||} \tag{13}$$

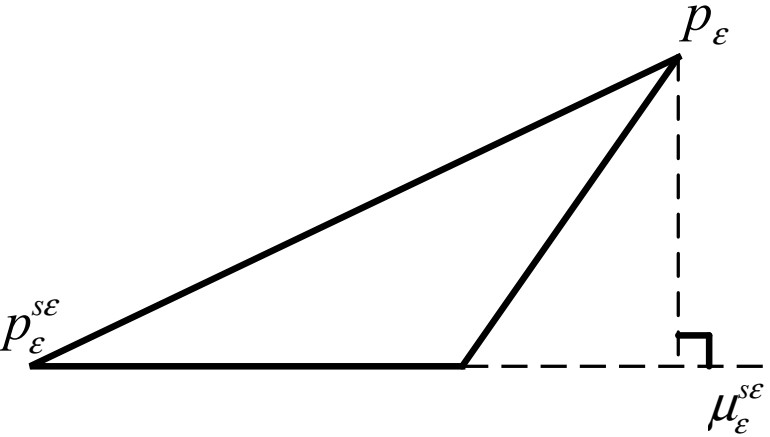

**Figure 4.** Illustration of the edge point-to-line residual. $u_\varepsilon^{sm}$ s the main direction of the line, and $p_s^{sm}$ is the geometric center of the five nearest points in the local map.

In the same way, each planar point in the current scan $p_s \in s_k$, we search for five points on the local map to form a plane. However, the difference is that the eigenvector corresponding to the minimum eigenvalue of the five-point covariance matrix is the normal vector corresponding to this plane. As shown in the Figure 5, $u_s^{sm}$ is the main direction of the normal vector. $p_s^{sm}$ is the geometric center of five planar points.

$$f_s(p_s) = (T_k p_s - p_s^{sm}) \bullet u_s^{sm} \tag{14}$$

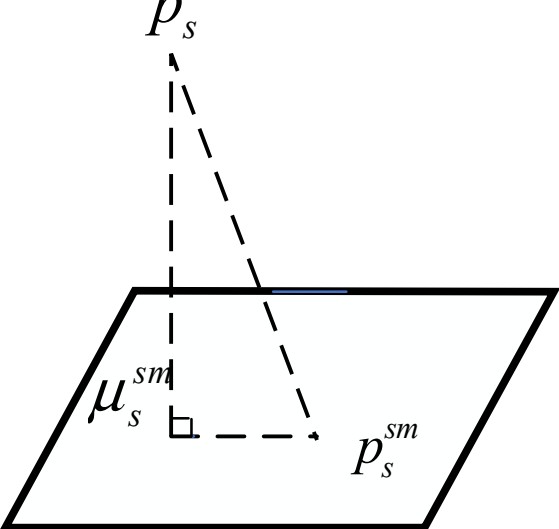

**Figure 5.** Illustration of the planar point-to-plane residual. $p_s^{sm}$ is the geometric center of five planar points. $u_s^{sm}$ is the main direction of the normal vector.

Therefore, this optimization problem can be constructed as:

$$\min\left\{\sum f_\varepsilon(p_\varepsilon) + \sum f_s(p_s)\right\} \tag{15}$$

The graph optimization algorithm is used to solve the nonlinear optimization problem. Thus, accurate state estimation can be obtained. Jacobian's derivation can be based on the mathematical model of left disturbance with $\delta\varphi \epsilon se(3)$ [25].

$$
\begin{aligned}
J_p &= \frac{\partial(TP)}{\partial\delta\varphi} \\
&= \lim_{\delta\varphi\to 0} \frac{(\exp(\delta\varphi)(TP) - (TP))}{\partial\varphi} \\
&= \begin{bmatrix} I_{3*3} & -[TP]_\times \\ 0_{1*3} & 0_{1*3} \end{bmatrix}
\end{aligned} \tag{16}
$$

where $[TP]_\times$ transforms 4D point expression $\{x, y, z, 1\}$ into 3D point expression $\{x, y, z\}$ and calculates its skew symmetric matrix. Jacobian matrix with edge residual can be derived by:

$$J_\varepsilon = \frac{\partial f_\varepsilon(p_\varepsilon)}{\partial(TP)} \frac{\partial(TP)}{\partial\delta\varphi} = p_n \bullet (u_\varepsilon^{sm} \times J_p) \tag{17}$$

In the same way, we also can derive:

$$J_s = \frac{\partial f_s(p_s)}{\partial(TP)} \frac{\partial(TP)}{\partial\delta\varphi} = u_\varepsilon^{sm} \bullet J_p \tag{18}$$

According to above formula, the estimation can be calculated by iterative optimization until it converges. In the work, the local map size is set to within 50 m radius. We propose the new optimization model, deduce the corresponding residual and Jacobian, and improve the solving speed significantly compared to other algorithms (see Section 6).

## 5. The Global Optimization Module

If the motion change in the current scan is greater than a certain threshold (10° in rotation and 0.5 m in translation) compared with that of the previous scan, the current frame will be judged as a key frame, and it will enter the global optimization which is based on sliding window.

In this paper, the state vector in the sliding window is defined as $\chi = [x_0, x_1, x_2, \ldots, x_n]$. And $\chi_i = \left[p_{b_i}^W, q_{b_i}^W, v_{b_i}^W, b_a, b_g\right]$. For the n keyframe window width, these states are obtained by minimizing

$$\underbrace{\min}_{x_n}\left\{||R_p\left(\tilde{\chi}\right)||^2 + \sum_{k=1}^{n} L\left(\hat{z}_{b_j}^{b_i}, \chi\right) + \sum_{k=1}^{n} \varkappa\left(\hat{z}_{b_j}^{b_i}, \chi\right) + \sum_{k=1}^{n} F\left(\hat{z}_{b_j}^{b_{loop}}, \chi\right)\right\} \tag{19}$$

In this formula, $R_p(\tilde{x})$ means the prior residual according to the measurements which are marginalized out because of the sliding window. $L(x_k)$, $\varkappa(x_k)$ and $F(x_k)$ denote the LIDAR, IMU and loop closure error terms. Figure 6 shows the optimization process.

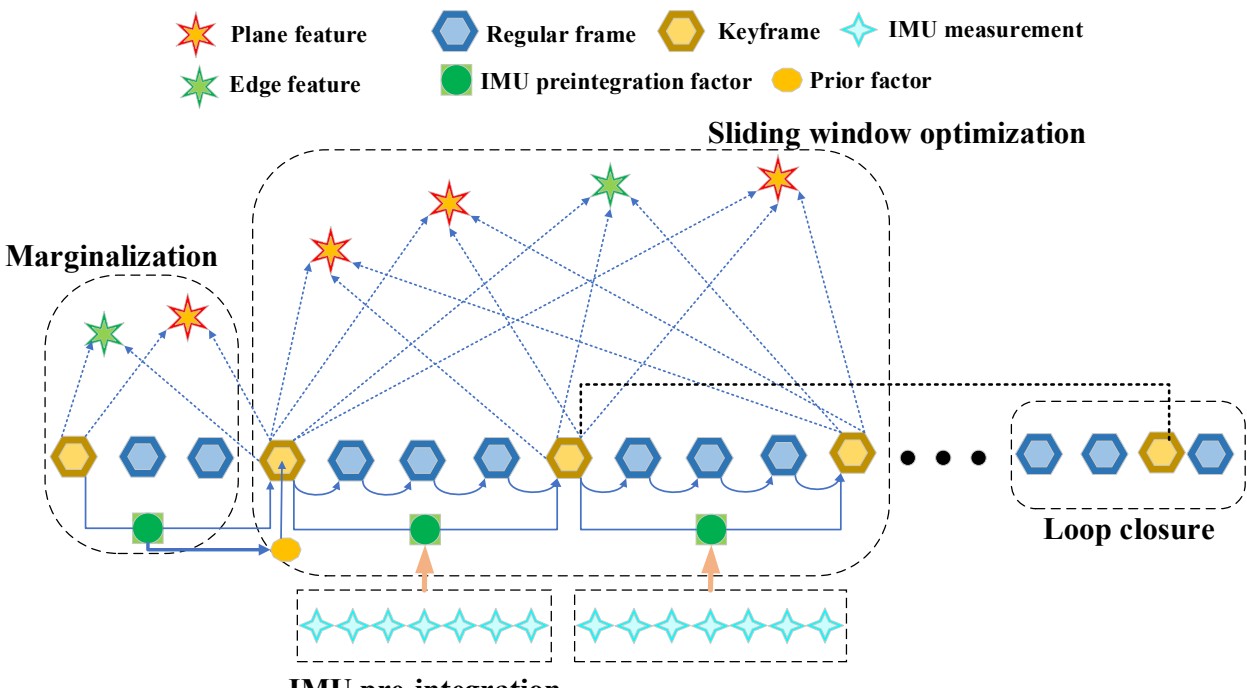

**Figure 6.** The optimization contains prior term, LIDAR term, IMU term and loop term. Prior term is generated by marginalization. Observations of LIDAR can provide scan-to-scan constraints and IMU pre-integration forms the constraints between keyframes. Loop closure is used for reduce the drift for the long-time running.

### 5.1. Prior Term

The purpose of marginalization is to bound the computational complexity. For the states out of the sliding window, they cannot be directly throwed away, because it will destroy the original constraint relationship and lose the constraint information. This work selectively marginalizes out $x_i$ from the sliding window via Schur-complement [26], and convert measurements corresponding to marginalized states into the prior.

### 5.2. LIDAR Term

Through the previous scan-to-map calculation of each scan (see Section 4.3), The LIDAR state variation between two adjacent frames is added into the graph optimization model as scan-to-scan constraint.

$$L\left(z_{b_j}^{b_i}, \chi\right) = \Delta T_{ij} = T_i^T T_j \tag{20}$$

This work assumes that $j$ and $i$ are the current and previous frame, respectively. This term can inhibit the accumulation of cumulative errors over a long time.

### 5.3. IMU Term

When IMU measurements are available, the residual between two continuous frames can be calculated, and the residual is defined as:

$$\varkappa\left(\hat{Z}_{b_j}^{b_i}, \chi\right) = \begin{bmatrix} \delta\alpha_{b_j}^{b_i} \\ \delta\beta_{b_j}^{b_i} \\ \delta\theta_{b_j}^{b_i} \\ \delta b^a \\ \delta b^\omega \end{bmatrix} = \begin{bmatrix} R_W^{b_i}\left(p_{b_j}^W - p_{b_i}^W - v_i^W \Delta t + \frac{1}{2}g^W \Delta t^2\right) - \hat{\alpha}_{b_j}^{b_i} \\ R_W^{b_i}\left(v_j^W - v_i^W \Delta t + g^W \Delta t\right) - \hat{\beta}_{b_j}^{b_i} \\ 2\left[\left(\hat{q}_{b_j}^{b_i}\right)^{-1} \otimes \left(q_{b_i}^\omega\right)^{-1} \otimes q_{b_j}^\omega\right]_{xyz} \\ b_j^a - b_i^a \\ b_j^\omega - b_i^\omega \end{bmatrix} \tag{21}$$

where $[\bullet]_{xyz}$ is the imaginary part of a quaternion. $\hat{\alpha}_{b_j}^{b_i}$, $\hat{\beta}_{b_j}^{b_i}$ and $\hat{q}_{b_j}^{b_i}$ are the pre-integration of position, velocity, and rotation between $j$ and $i$ under the assumption that $b^a$ and $b^\omega$ are stable.

### 5.4. Loop Term

Loop closure is an important step to correct the accumulated error in SLAM system. In this study, the function is realized by distance detection. In the current frame, we search for the distance coordinates of nearby key frames. And frames within the geometric radius 15m can be marked as the candidate loop closure frames.

We select the nearest frame from the candidate frame as the previous key frame $T_{loop}^W$. Then a certain number of point clouds are found near the previous key frame, which are used for a small local map. In this module, the number of points in optimization model is less, so ICP is used to calculate the relative transformation $T_{W'}^W$ of similar scenes. The residual between the previous key frame and current frame can be obtained:

$$F\left(\hat{z}_{b_j}^{b_{loop}}, \chi\right) = \Delta T_{loop,j} = \left(T_{loop}^W\right)^{-1} T_{W'}^W T_j^{W'} \tag{22}$$

In order not to affect the real-time performance, loop closure detection and mapping for ICP are in another thread.

## 6. Evaluation

In order to verify our algorithm, we have conducted public dataset experiments and real-word experiments. The proposed algorithm is operated on a laptop which consists of an Intel-i7 CPU and 16G of memory. The operating system is Ubuntu18.04 and ROS Melodic [27]. We use evo [28] to evaluate accuracy. The optimization library we used are GTSAM [29] and Ceres [30].

### 6.1. Validation of M2DGR Datasets

M2DGR [31] datasets were recorded using ground robots. As shown in Figure 7, A HDL 32E Velodyne LiDAR (labeled 3 in the figure) was used to scan the surrounding environment and obtain the 3D point cloud. The IMU device is Handsfree A9 (labeled 5 in the figure), which is a 9-axis sensor. In outdoors, the satellite visibility is good so that the GNSS-RTK suite (labeled 4 in the figure) outputs high-precision ground truth. For indoor environments, the ground truth trajectories are recorded with a motion-capture system which consists of twelve highspeed tracking cameras. The spatial relationship among different sensors have been calibrated.

In order to test the robustness of our algorithm, we adopt tests of different scenarios, and the data information is shown in the Table 1. Street_01, 04, 07, 10 are collected on the street. In the street dataset, there are buildings discontinuously. The structured environment has rich geometric feature information. However, the switching of unstructured scene has the unpredictable influence on LIDAR odometry. Various weeds, leaves and other environmental factors affect the positioning accuracy in Street_04 around the lawn. Loop is set for loop closure detection, which is important for the validation in back-end graph optimization, and the motion state of zigzag brings challenges for interframe motion estimation. When the running time is longer than 500 s, we think that it is long-term to test for stability and robustness. Gate_02 is collected around the large circular gate. It is easy to satisfy the loopback condition. The ground robot is always rotating in Circle_02 scene, which is difficult for the feature matching, and Hall_05 is collected for indoor environment. There is a large amount of overlap and structured feature during the experiment.

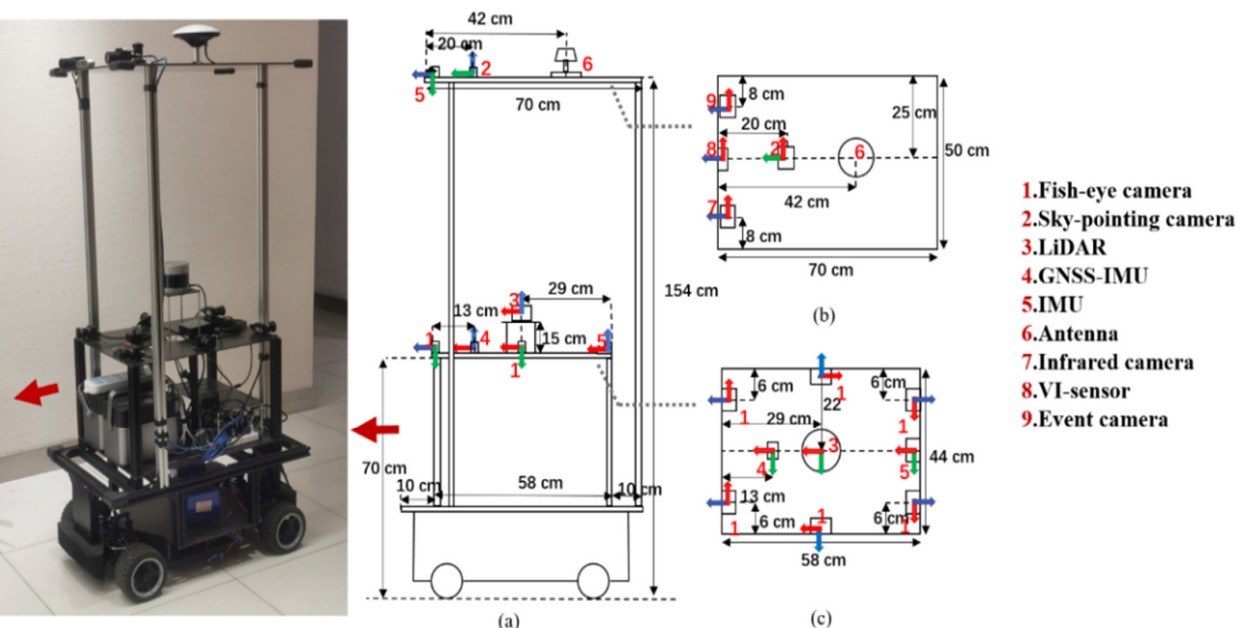

**Figure 7.** The sensor suite of M2DGR. Sensors are strictly calibrated and using the same time stamps.

**Table 1.** The dataset contains all kinds of scene.

|  | Street_01 | Street_04 | Street_07 | Street_10 | Gate_02 | Circle_02 | Hall_05 |
|---|---|---|---|---|---|---|---|
| Durations/s | 1028 | 858 | 929 | 910 | 327 | 244 | 402 |
| Description of features | Street and buildings, zigzag, long-term | Around lawn, loop back, long-term | Zigzag, long-term | Zigzag, long-term | Loop back, around gate | Circle, rotation | Indoor, large overlap |

We contrast our system with ALOAM, LeGo-LOAM and LIO-SAM. ALOAM only depends on LIDAR; the core of the algorithm is the same as LOAM, but it is achieved according to Ceres for the code readability. LeGo-LOAM uses IMU data to help remove motion distortions from point clouds. LIO-SAM is a tightly coupled LIDAR and IMU approach, but there is no front-end processing of point clouds and the traditional solution algorithm in scan-to-map is more time consuming.

We choose some typical scenarios such as zigzag, rotation and loop. In these cases, point cloud mismatching often occurs due to the violent motion of the carrier. The long-term run is to verify the elimination of the accumulated error of the LIO system.

In Table 2, the bold and italic values indicate the minimum error. Seven groups of experiments prove that our algorithm improves the accuracy in most scenarios. Especially in a scene such as a street. In the sequence "Circle_02", our LIO system has a higher error than LeGo-LOAM. That is because "Circle_02" is collected in a fixed scene and the ground robot is always rotating. This motion state has slightly bad effect on IMU pre-integration. Other than "Circle_02", our system benefits from the tightly couple of inertial and LIDAR information. In the sequence "Gate_02" and "Hall_05", the four algorithms perform equally well. These two scenes are simple and rich in structural features. However, in the street sequence, our algorithm can greatly improve the performance.

**Table 2.** Absolute trajectory error (ATE) RMSE (m) of the four algorithms in seven datasets.

|            | Street_01 | Street_04 | Street_07 | Street_10 | Gate_02 | Circle_02 | Hall_05 |
|------------|-----------|-----------|-----------|-----------|---------|-----------|---------|
| ALOAM      | 7.661     | 3.582     | 27.590    | 22.075    | 0.361   | 1.391     | 1.029   |
| LeGo-LOAM  | 3.269     | 1.193     | 14.583    | 31.024    | 0.485   | *0.288*   | 1.034   |
| LIO-SAM    | 6.390     | 1.133     | 4.693     | 2.569     | 0.326   | 0.618     | 1.053   |
| OURS       | *1.362*   | *0.836*   | *1.579*   | *1.479*   | *0.313* | 0.409     | *0.980* |

### 6.1.1. Positioning Performance Analysis

Street_01 is chosen for our analysis. Figure 8 shows the trajectories of the four algorithms in street_01 in X-Y plot. The accumulative error of four algorithms can be obtained from the detail diagram. Our system makes reasonable use of feature information, which effectively improve the accuracy of point cloud matching. Planar points are extracted from ground surface feature information, and edge points are extracted from rod feature information. We notice that ALOAM, LeGo-LOAM and LIO-SAM will drift and have a large deviation when it comes to turning. However, in this system, the point cloud registration has been greatly improved.

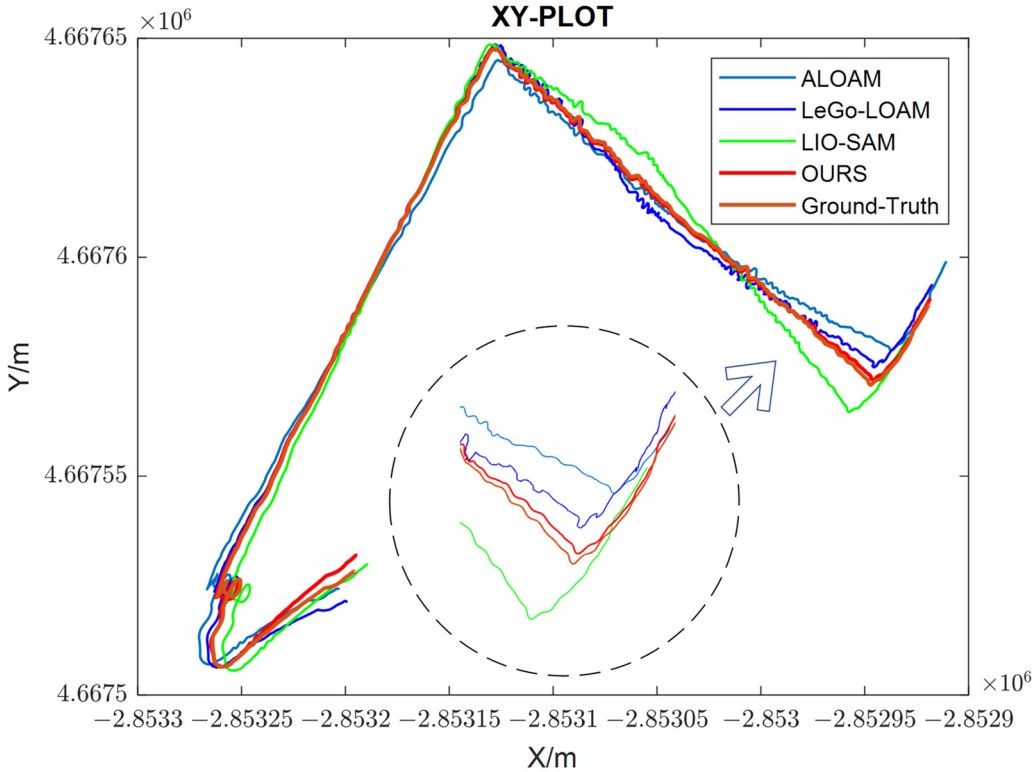

**Figure 8.** The trajectories of the four algorithms on street_01. Our trajectory is closest to the ground truth. The detail diagram is near the end of this test.

More detailed evaluations have been conducted. Figure 9 shows absolute trajectory error variation for 4 algorithms in street_01. ALOAM have largest error without IMU measurements, and its scan-to-scan method has a bad influence, which is easy for providing inaccurate information in the scan-to-map module. LeGo-LOAM applies a two-step scan-to-scan method, which is beneficial for improving efficiency, but it still introduces much error. Moreover, without IMU constraint, loose-coupled LIO system such as LeGo-LOAM cannot adequately make use of sensor observation information. In LIO-SAM, there is no point cloud preprocessing section. Lots of unstable observations also have bad effects on point cloud registration.

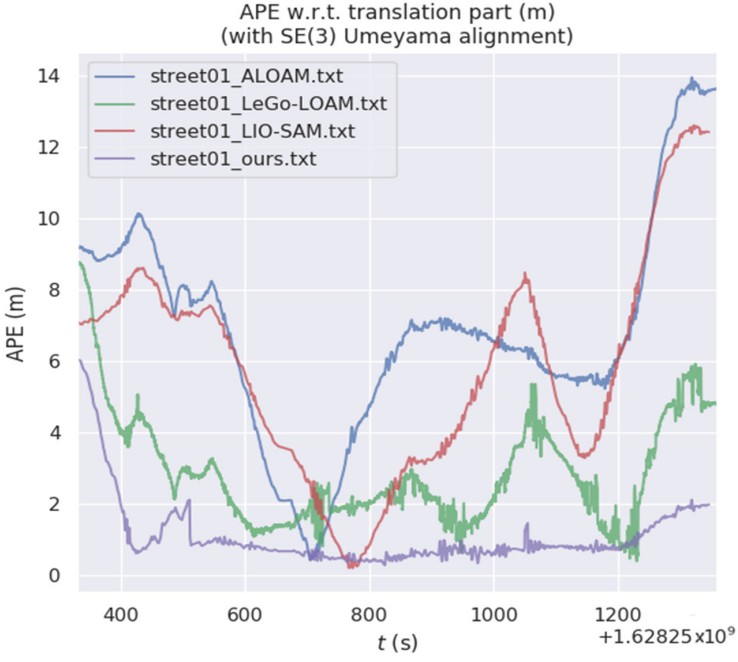

**Figure 9.** Illustration of the absolute trajectory error variation. Our system has been kept a low level.

This work not only tests RMSE in Table 2. Figure 10 displays each evaluation parameter. Our system has a good performance in different indicators. Figure 11 is the box diagram, which is used to display the dispersion of a set of data. The system also has the lowest deviation.

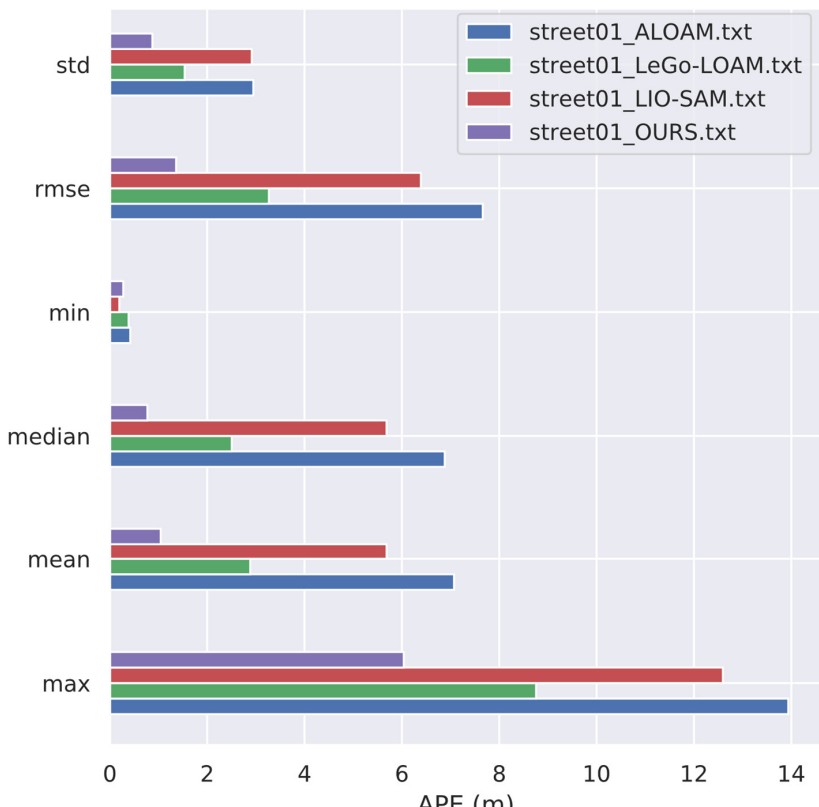

**Figure 10.** Illustration of A variety of indicators. The superiority of our algorithm can be concluded from the statistical graphs.

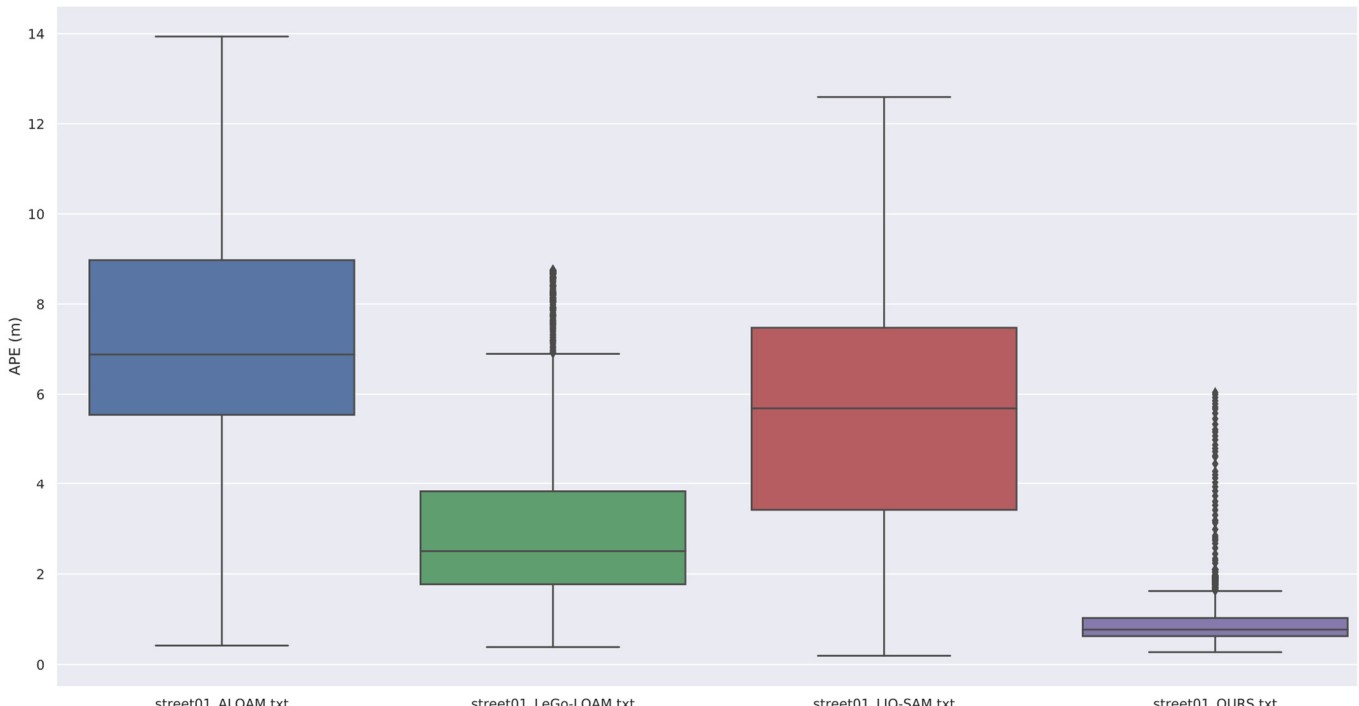

**Figure 11.** Box diagram which is used to reflect the characteristics of distribution of data.

Four algorithms have been compared. We now analyze the difference between our system and ground truth. Figure 12 is the display diagram of trajectory and truth value in the X-Y plane.

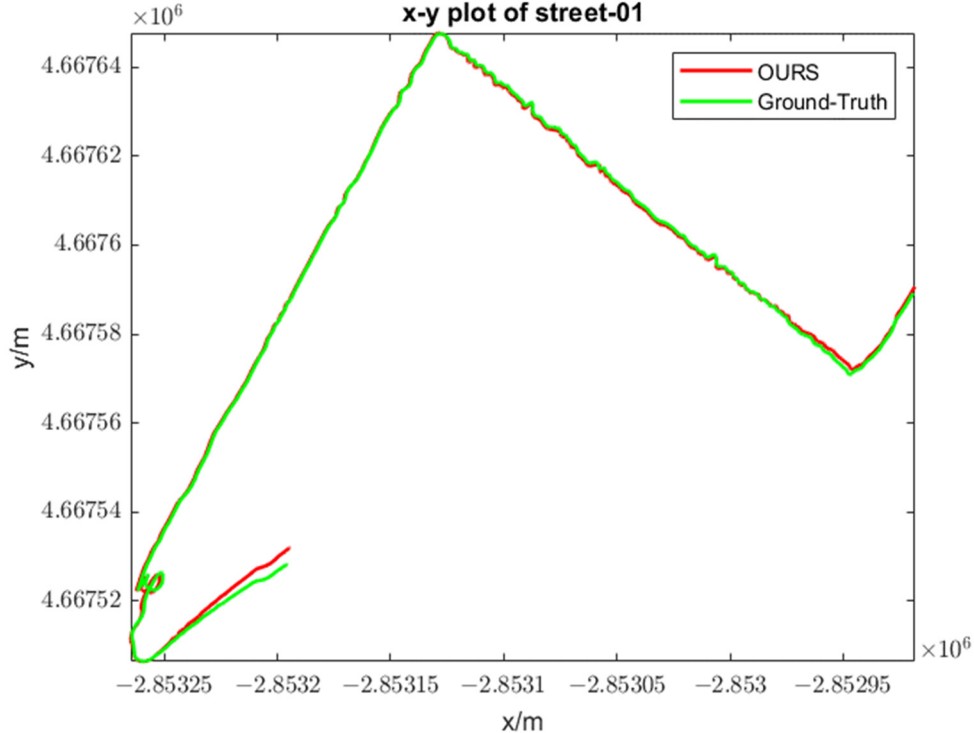

**Figure 12.** The detailed trajectories of ground truth and our system in street_01. They are aligned well.

Figure 13 shows error changes in three directions throughout the period. We can see that the system has an obvious deviation at the start time. That is because the optimization process takes time to converge and correct. At the same time, local map takes time to build.

After a long run, the error is remained low in our system. It verifies the robustness and high precision of the work.

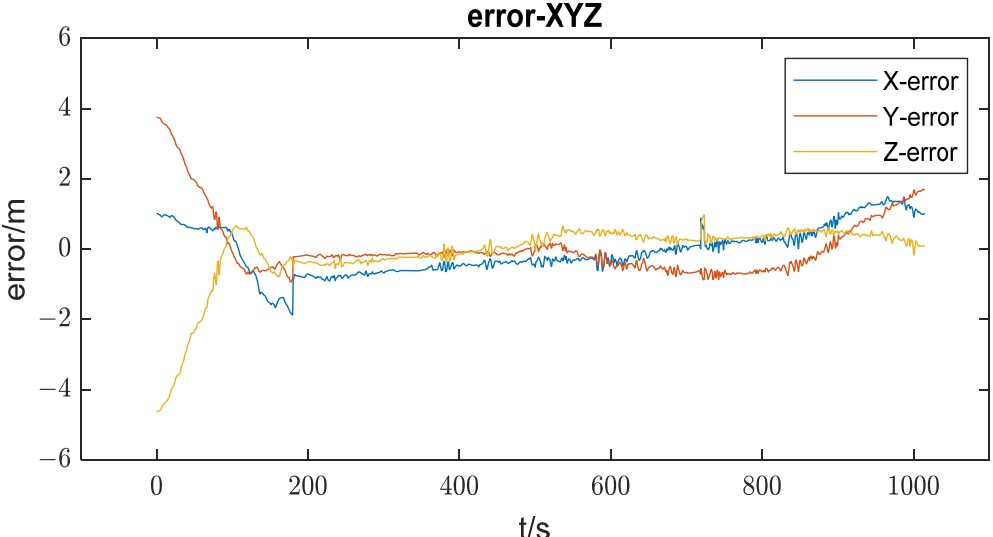

**Figure 13.** Error changes in three directions. The initial error is a little larger, after that the error is small.

### 6.1.2. Runtime Performance Analysis

Meanwhile, our experiments proved that the time consumption of our scan-to-map module is significantly reduced from Table 3. Four algorithms have this module which is the most time-consuming. So, we choose the cost time of this module for comparison. The bold and italic values indicate the minimum time consumption. We can see the obvious advantage of our algorithm.

**Table 3.** The time consumption (ms) in seven datasets. All are recorded in the same platform.

|  | Street_01 | Street_04 | Street_07 | Street_10 | Gate_02 | Circle_02 | Hall_05 |
|---|---|---|---|---|---|---|---|
| ALOAM | 295.875 | 250.093 | 309.164 | 251.751 | 230.516 | 294.688 | 106.374 |
| LeGo-LOAM | 132.986 | 93.184 | 149.694 | 124.700 | 112.126 | 122.932 | 57.938 |
| LIO-SAM | 61.900 | 43.461 | 83.957 | 76.541 | 57.518 | 90.450 | 21.308 |
| OURS | *37.600* | *27.427* | *66.093* | *50.888* | *35.215* | *34.628* | *14.296* |

LOAM and LeGo-LOAM use a scan-to-scan match to provide odometry, which means using the current scan and last scan to do the scan matching, and the result offers an initial guess for mapping. LIO-SAM and our system use IMU pre-integration, which is of high frequency, and we use the back-end result to suppress IMU drift. Even more, thanks to the edge points extracted from the rod-shaped information and the planar points extracted from the ground surface information, many outliers are not in the operation. Accuracy and speed are greatly improved.

We still choose street_01 for analysis. Table 4 shows the number of frames four systems processed.

**Table 4.** The scan-to-map frames in street_01.

|  | ALOAM | LeGo-LOAM | LIO-SAM | OURS |
|---|---|---|---|---|
| Scan-to-map frames | 2788 | 2566 | 5128 | 5133 |

Figure 14 shows the processing time of each frame. We can clearly see the lowest cost time of our system. In ALOAM, the mapping module uses the global map and map maintenance is time-consuming. LeGo-LOAM and LIO-SAM are the same, which apply Levenberg–Marquardt algorithm [32] of 30 iterations for optimization. Our algorithm uses faster and more accurate graph optimization model to solve the scan-to-map module (see Sections 4.2 and 4.3).

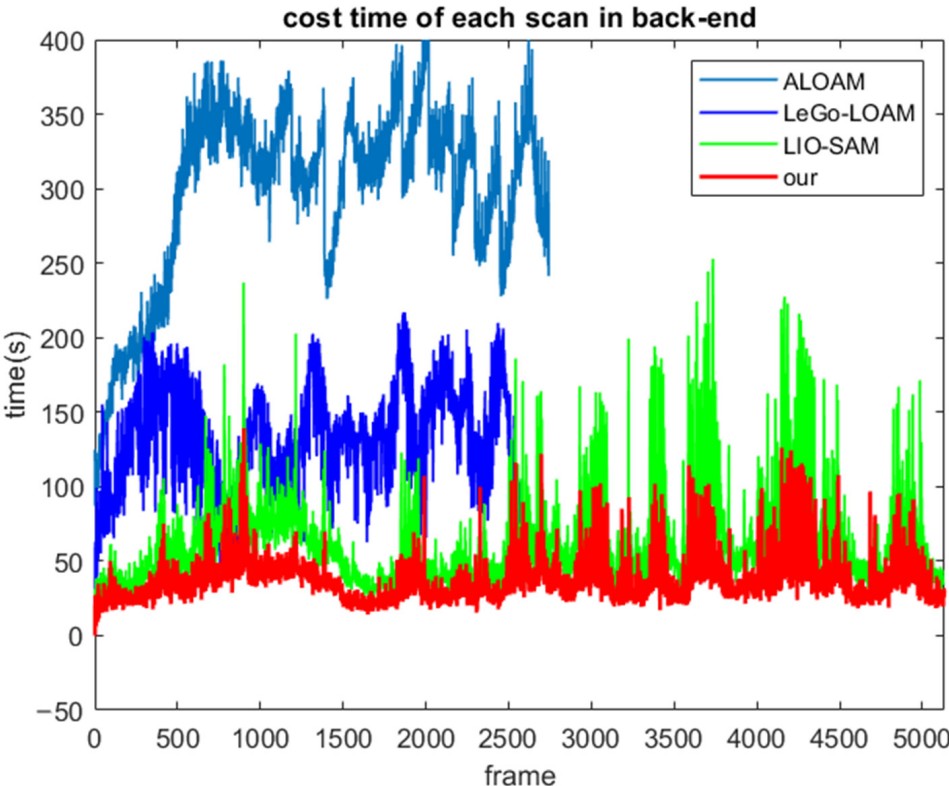

**Figure 14.** Processing time of each scan. The red is obviously lower than other three.

The computationally-efficient system is meaningful for mobile terminal and other platforms with limited computing resources.

### 6.2. Validation of Our Datasets

To further test our system, LIDAR has less beams and IMU is of different quality. We set up a sensor suite composed of a VLP16 Velodyne and an ADIS16488 IMU (see Figure 15). Sensors have hardware time synchronization because of GPS pulse per second (PPS). RTK/IMU combined navigation results are used as the truth value, which is after NovAtel Inertial Explorer software post-processing. Our aim is to prove the versatility of our algorithm. We pick two typical scenarios. One is in a campus (dataset_01) and the other is on a city road (datasetet_02).

In dataset_01, (see Figure 16) the speed of our car is about 6 m/s. There is rich feature, but there is accumulated error in long-term run. Various weeds, leaves and other environmental factors affect the positioning accuracy.

Dataset_02 was collected in the wide urban road (see Figure 17) and the speed of our car is about 14 m/s. It contains a large number of buildings. Dynamic objects will affect the point cloud matching accuracy. The density of point cloud in open space is small and it is difficult to have a good performance on localization.

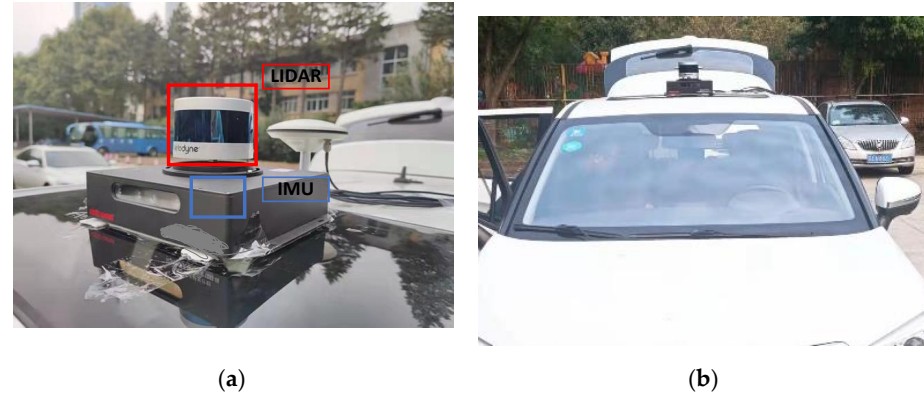

(**a**)　　　　　　　　　　　　　　　(**b**)

**Figure 15.** Our sensor suite. All have hardware time synchronization (**a**) IMU is in the LIDAR below. Integrated navigation device is used to gain ground truth. (**b**) The device is mounted on top of the vehicle.

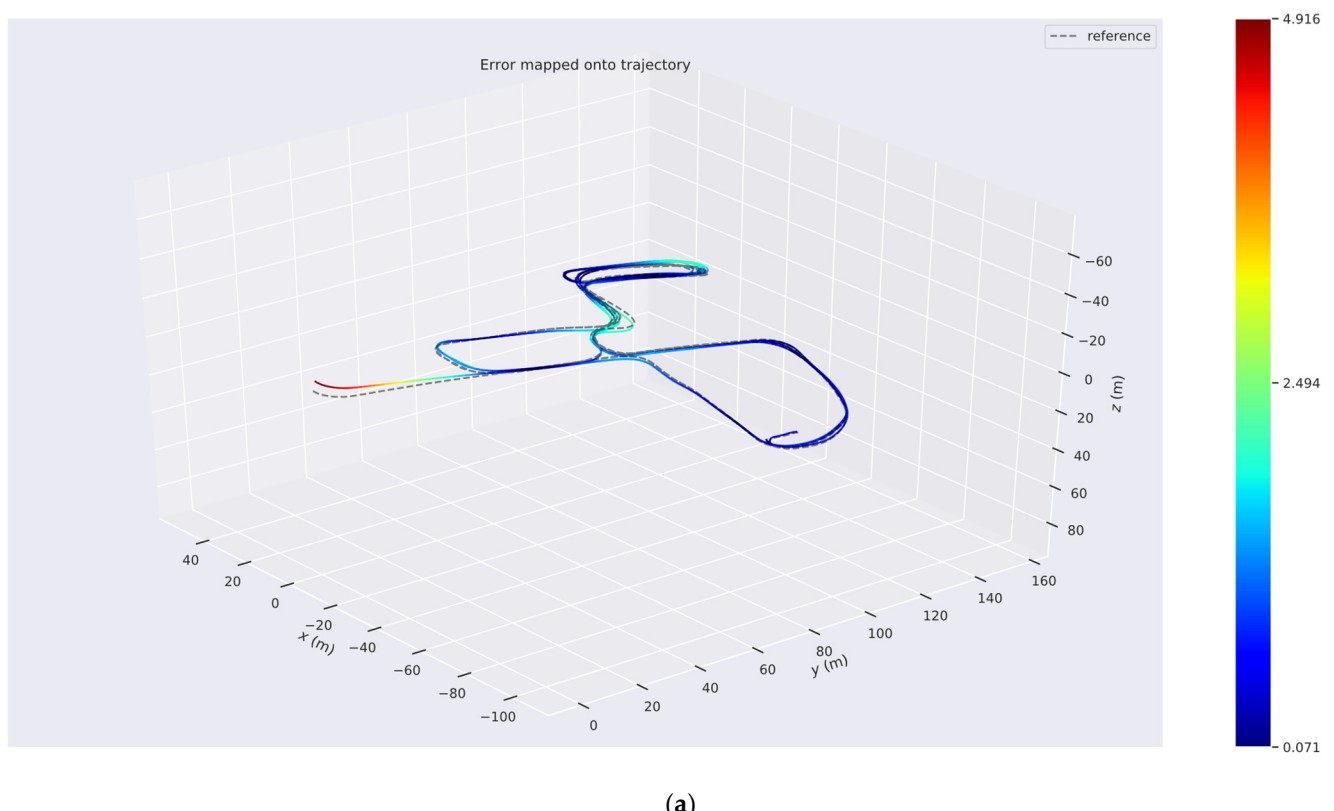

(**a**)

**Figure 16.** *Cont.*

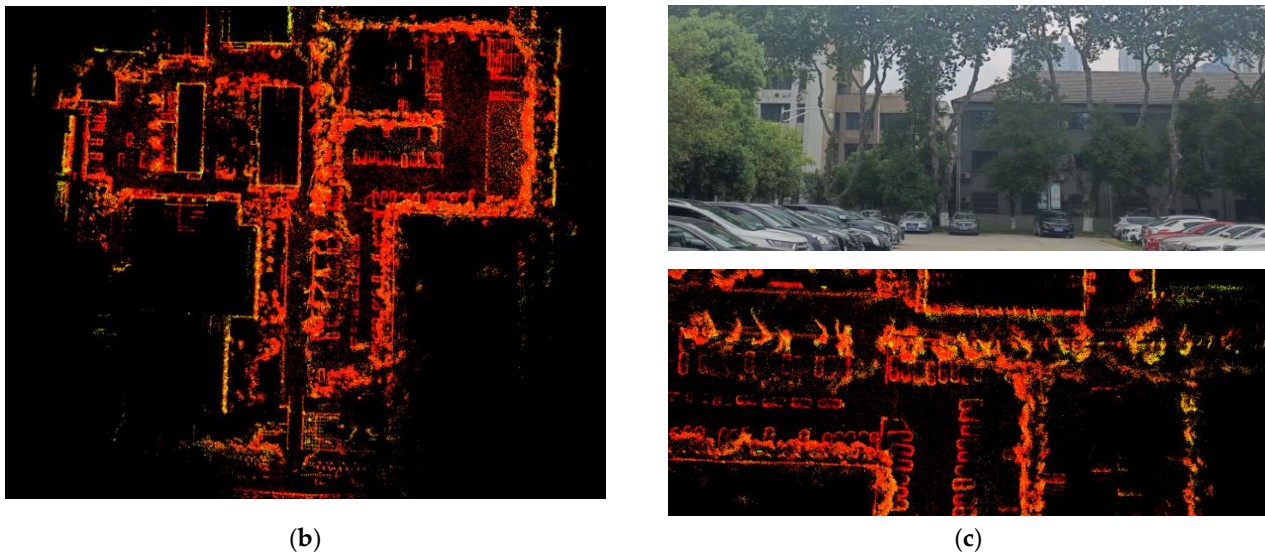

(**b**)                                                                                                 (**c**)

**Figure 16.** Trajectory and mapping are generated by our system. (**a**) Our trajectory and ground truth. Different colors represent the error values. (**b**) The mapping result is rendered with LIDAR intensity value from the top view during the positioning process. (**c**) The top panel is the specific real-word environment picked out of the whole trajectory. The bottom panel shows the detail from LIDAR mapping correspond to the top panel.

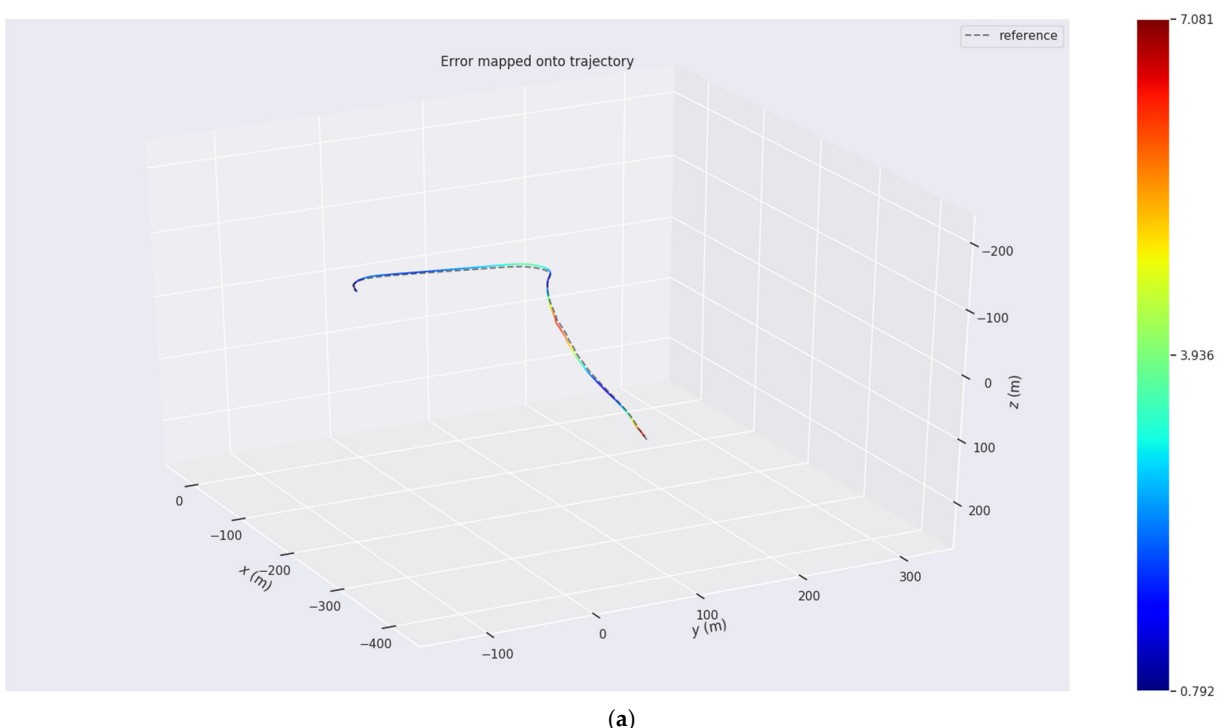

(**a**)

**Figure 17.** *Cont.*

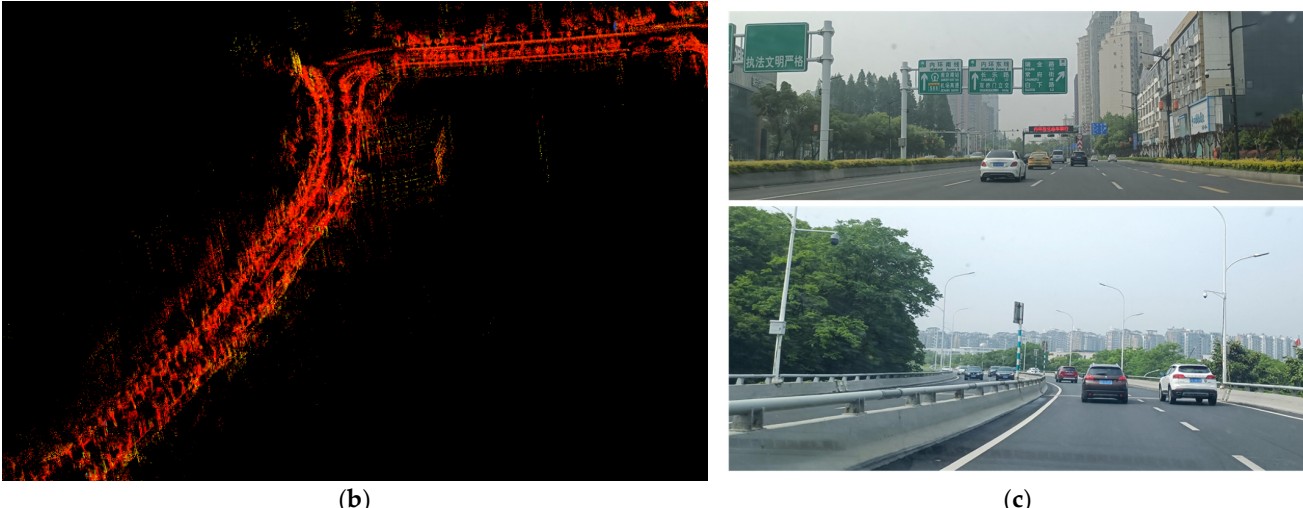

**(b)** 　　　　　　　　　　　　　　　　　　　　　　　**(c)**

**Figure 17.** Trajectory and mapping in dataset_02. (**a**) Our trajectory and ground truth in the urban road. (**b**) The mapping result is rendered with LIDAR intensity value from bird-eye view. (**c**) The road scene.

In feature-rich areas(dataset_01), we can conclude that LIDAR will have good performance than the wide-open spaces(dataset_02) from the Table 5. The bold and italic values indicate the minimum error. Compared with the LIO-SAM, the RMSE in the study increases by 24.2% in dataset_01 and 25.0% in dataset_02.

**Table 5.** Absolute trajectory error (ATE) RMSE (m) in our real-word experiments.

|  | Durations/s | Description of Features | A-LOAM | LeGo-LOAM | LIO-SAM | OURS |
|---|---|---|---|---|---|---|
| dataset_01 | 678 | campus | 4.040 | 1.264 | 1.272 | *0.964* |
| dataset_02 | 240 | wide rode | 6.391 | 5.933 | 4.205 | *3.152* |

In the Figures 16a and 17a, they show the corresponding trajectories of the two datasets and different colors represent the error values. In the Figures 16b and 17b, the global reconstruction of the two scenes is built. Due to the multiple constraints of the back-end optimization, we obtain a globally consistent point cloud map. According to Equation (19), the robustness and reliability of the map can be guaranteed. The map shows the structural details in the bottom panel of Figure 16c. We can clearly see the cars and the trunk of the tree in the dataset in bird-eye view. In the Figure 17c, the wide roads have great influence for LIDAR slam (see Table 5). However, our algorithm can also reduce the error.

The time performance is consistent with the M2DGR dataset analysis (see Section 6.1.2). In dataset_01, due to the features' richness in campus scenes, the feature information relationship in scan-to-map needs much time for calculation. However, the average time consumption is 64.379 ms, which can satisfy the real-time requirement (LIDAR is 10 HZ sampling frequency), and the average time consumption is 25.874 ms in the road test in the dataset_02, which is computationally efficient. Our system can achieve a good estimation result with less time cost.

## 7. Conclusions and Future Perspectives

According to the datasets and our own data experiments, compared to LIDAR only positioning (ALOAM), the positioning accuracy and robustness is significantly improved. Then only IMU data helps the point cloud to remove distortion (LeGo-LOAM), the tightly coupled LIO has lower drift, and compared to LIO-SAM, segmentation and clustering are used to mark feature information. The point cloud matching is more accurate and the runtime of scan-to-map module is much less.

In this paper, we propose an improved LIO system. Firstly, it makes reasonable use of the feature information of point cloud and effectively improves the accuracy of point cloud matching. Point cloud registration is carried out after marking rod-shaped and planar feature information which is different from the existing LIDAR-inertial integration scheme. The optimized edge points and planar points extraction modes reduce the computation of scan-to-map and improve the real-time performance. Secondly, prediction of IMU odometry and correction of LIDAR odometry improve the accuracy and frequency of the mapping module, which is inspired by LIO-SAM. Comparing this to the front-end odometry in traditional scan-to-scan mode, the tightly coupled mode of system greatly improves the performance of LIO. Thirdly, the scan-to-map based on the graph optimization model is of great significance to speed up the solution and decrease error. Therefore, the system does not apply the Levenberg–Marquardt algorithm, which is adopted in Lego-LOAM and LIO-SAM. Fourthly, the robust back-end optimization system including effective loop closure suppress the cumulative drift of LIO odometry, and IMU measurements residuals add more constraints information between IMU and LIDAR measurements compared to LIO-SAM. The optimization mode based on sliding window ensure full use of sensors information under real-time conditions. Experiments show that the real-time performance and accuracy of our algorithm exceed that of most state-of-the-art systems in various typical environments.

It can be seen from Table 2 that the positioning accuracy (RMSE) can be improved by 25–78% (the average increment is 64.45%) in the M2DGR street datasets compared to the current tightly coupled LIDAR SLAM algorithms (LIO-SAM). After optimizing the extraction mode of edge points and planar points, our system processes more frames and takes less time on average, effectively improving real-time performance. In our actual scene datasets, the RMSE in the study increases by 24.4% in dataset_01 and 25.0% in dataset_02.

We draw a conclusion that we propose the low drift and high real-time LIDAR-inertial positioning and mapping system, which is of great importance in indoor locating and other GNSS occlusion area. At the same time, it can provide high precision point cloud image for scene understanding in automatic system. For the back-end optimization framework, we can easily add other measurements such as GNSS for global restriction.

In the future, we noticed that it is necessary to improve the initialization process to reduce initial error. It is very important to judge the rod-shaped feature information and the planar feature information in the research process of this paper. This work gets thresholds according to experience temporarily. We will focus on online threshold estimation and adaptive threshold selection. Also, it is worth mentioning that LIO system is prone to Z direction drift in the large scene. Then more constraints will be introduced to suppress drift in our next step. Furthermore, according to the recent study [33–36], the positioning and mapping system based on solid state LIDAR can significantly reduce the hardware cost. Therefore, the research of solid-state LIDAR- inertial system is worth exploring.

**Author Contributions:** Conceptualization, H.L., S.P., W.G. and C.M.; methodology, H.L.; software, F.J. and C.M.; validation, S.P. and F.J.; formal analysis, H.L. and C.M.; investigation, W.G. and F.J.; resources, S.P., W.G. and H.L.; writing—original draft preparation, H.L. and X.L.; writing—review and editing, W.G. and X.L.; supervision, S.P. and W.G. All authors have read and agreed to the published version of the manuscript.

**Funding:** This research study was funded by the National Key Research and Development Program of China (No. 2021YFB3900804), the Research Fund of Ministry of Education of China and China Mobile, (No. MCM20200J01) and the Fundamental Research Funds for the Central Universities (No. 2242021R41134).

**Data Availability Statement:** Not applicable.

**Conflicts of Interest:** The authors declare no conflict of interest.

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
