# Peer review of "LIDAR-Inertial Real-Time State Estimator with Rod-Shaped and Planar Feature"

_remotesensing, doi:10.3390/rs14164031_

Round 1

Reviewer 1 Report

The algorithm for real time automated point cloud registration is formulated and presented well. The literature review, mathematical formaulation and then execution of the algorithm with real world data showed good performance, there I recommend to publish. The comparison with some existing algorithms shown significant improvement in the proposed method.

Author Response

 Thank you for the reviewer’s recognition! We gratefully appreciate for your valuable comment. We will work harder in future research.

Reviewer 2 Report

The authors have developed a very effective LIDAR-Inertial real-time state estimator with rod-shaped and planar feature that

are important for autonomous system. It is rightly stated that a novel point cloud feature selection for LIDAR-inertial tightly coupled system may improve its quality because the preprocessing method reduces the outliers. From the results presented, it can be concluded that the proposed system outperforms much higher positioning accuracy than the state-of-the-art methods in various scenarios. Despite its many advantages, I noticed some minor errors in the manuscript that needed to be corrected.

1. For the sake of the reader, each acronym should be clarified (expanded) at its first appearance.

2. Lines 111-119: The margin on the left side of the text should be corrected.

3. Variables (lines 143-148) should be written in italics.

4. Line 305: 5.1 prior term => 5.1 Prior term

5. For all citations, enter a space: evo[26] => evo [26]

6. Lines 466-468: The caption under the figure should be on the same page;

7. As the manuscript is for the readers of Remote Sensing, 2-3 sources from this journal should be cited in order to take into account what has already appeared in this journal.

Reviewer 3 Report

With interest we were reading the paper and we have the following comment:

1. On line 508 the authors state that the current system is an improved version of the LIO-SAM. But it is not clear what is new and what is from the existing system.

2.The new system is contrasted with ALOAM, LeGo-LOAM and LIO-SAM but these systems are hardly referenced and not described. 

3. On different places different experimental results are reported. On line 477 is stated that RMSE has probably 25% increase both datasets?

4. On lone 56 the authors mention the KITTI benchmark site. But this benchmark has not been used but M2DGR datasets. Recently other publications report better experimental results. The choice of the current benchmark systems is not motivated.

LOCUS: A Multi-Sensor Lidar-Centric Solution for High-Precision Odometry and 3D Mapping in Real-Time. Matteo Palieri1,2 , Benjamin Morrell1 , Abhishek Thakur3 , Kamak Ebadi1 , Jeremy Nash1 , Arghya Chatterjee4 , Christoforos Kanellakis5 , Luca Carlone6 , Cataldo Guaragnella2 , Ali-akbar Agha-mohammadi1  

5. Test set. In line 352 the authors report test scenarios in global terms. To repeat/verify the experiments more info is needed.

6. fig 16 is difficult to understand

7. In section 6.2 the authors report about a real life traffic test set. We were wondering if this test set has been published?

8. The statement in line 499 needs more explanation.
